# Unique features of KGN granulosa-like tumour cells in the regulation of steroidogenic and antioxidant genes

**Feng Tang, Katja Hummitzsch, Raymond J. Rodgers** *

School of Biomedicine, Robinson Research Institute, The University of Adelaide, Adelaide, SA, Australia

* ray.rodgers@adelaide.edu.au

**Data Availability Statement:** The Supporting Information show the raw data from all the RNA-seq analyses. Original RNA-Seq files can be found at Gene Expression Omnibus (GEO) database under codes GSE161341 (sample GSM4905063),

## Abstract

The ovarian KGN granulosa-like tumour cell line is commonly used as a model for human granulosa cells, especially since it produces steroid hormones. To explore this further, we identified genes that were differentially expressed by KGN cells compared to primary human granulosa cells using three public RNA sequence datasets. Of significance, we identified that the expression of the antioxidant gene *TXNRD1* (thioredoxin reductase 1) was extremely high in KGN cells. This is ominous since cytochrome P450 enzymes leak electrons and produce reactive oxygen species during the biosynthesis of steroid hormones. Gene Ontology (GO) analysis identified steroid biosynthetic and cholesterol metabolic processes were more active in primary granulosa cells, whilst in KGN cells, DNA processing, chromosome segregation and kinetochore pathways were more prominent. Expression of cytochrome P450 cholesterol side-chain cleavage (*CYP11A1*) and cytochrome P450 aromatase (*CYP19A1*), which are important for the biosynthesis of the steroid hormones progesterone and oestrogen, plus their electron transport chain members (*FDXR, FDX1, POR*) were measured in cultured KGN cells. KGN cells were treated with 1 mM dibutyryl cAMP (dbcAMP) or 10 μM forskolin, with or without siRNA knockdown of *TXNRD1*. We also examined expression of antioxidant genes, $H_2O_2$ production by Amplex Red assay and DNA damage by γH2Ax staining. Significant increases in *CYP11A1* and *CYP19A1* were observed by either dbcAMP or forskolin treatments. However, no significant changes in $H_2O_2$ levels or DNA damage were found. Knockdown of expression of *TXNRD1* by siRNA blocked the stimulation of expression of *CYP11A1* and *CYP19A1* by dbcAMP. Thus, with *TXNRD1* playing such a pivotal role in steroidogenesis in the KGN cells and it being so highly overexpressed, we conclude that KGN cells might not be the most appropriate model of primary granulosa cells for studying the interplay between ovarian steroidogenesis, reactive oxygen species and antioxidants.

## Introduction

Primary human ovarian granulosa cells produce the steroid hormones progesterone and oestrogen. The production of oestradiol and progesterone in granulosa cells depends on

GSE130664 (samples GSM4162518 and GSM4162519) and GSE193123 (samples GSM5773736, GSM5773737 and GSM5773738). Any other data that support the other findings of this study are available from the corresponding author upon request.

**Funding:** The author(s) received no specific funding for this work.

**Competing interests:** The authors have declared that no competing interests exist.

expression of cytochrome P450 enzymes, namely P450 cholesterol side-chain cleavage (encoded by *CYP11A1*) and P450 aromatase (encoded by *CYP19A1*). Studies of primary granulosa cells, especially human cells, have been enhanced by the establishment of cell lines. The human granulosa-like tumour cell line KGN was originally established in 2001 by Nishi and colleagues from a patient with invasive ovarian granulosa cell carcinoma [1]. It has since been used widely as a model for primary ovarian follicular granulosa cells [1]. Compared to the five human ovarian granulosa cell lines developed before 2001 [2–6], KGN cells not only retain most of the physiological activities of granulosa cells but also express the follicle-stimulating hormone (FSH) receptor, like primary granulosa cells do [1]. Their ability to produce steroid hormones was confirmed by increased *CYP19A1* expression and production of oestradiol and progesterone after stimulation with either FSH, forskolin or dibutyryl cAMP (dbcAMP) by regulating adenylyl cyclase-cAMP-protein kinase A (PKA) signalling cascade [1, 7, 8]. However, the basal activity of CYP19A1 in KGN cells was only $0.84 \pm 0.23$ pmol/mg protein, which is about 50 times less than primary human granulosa cells where the CYP19A1 activity was $41.5 \pm 8.3$ pmol/mg protein [1, 9]. Nevertheless, the mechanism of steroidogenesis in KGN cells is quite similar to human granulosa cells. However, such intrinsically low activity of CYP19A1 in KGN cells may affect the cells' behaviours.

The steroidogenic enzymes CYP19A1 and CYP11A1 rely on electron transport chains for their activity [10]. This potentially results in the production of reactive oxygen species (ROS) due to electron leakage. A previous study showed that electrons were transferred from NADPH to CYP11A1 at approximate 85% during their catalytic function of steroid hydroxylation [11]. Hence 15% leaked and possibly formed ROS. To counteract ROS, granulosa cells normally express antioxidant enzymes. Antioxidant genes include glutathione peroxidases (GPX1-8), superoxide dismutases (SOD1/2), catalase (CAT), peroxiredoxins (PRDX1-6), glutathione disulphide reductase (GSR), thioredoxins (TXN and TXN2) and thioredoxin reductases (TXNRD1- 3). Deficiencies in antioxidants such as the mitochondrial Sod2 have resulted in decreased progesterone and oestradiol production due to decreased expression of StAR, the protein responsible for cholesterol transport into mitochondria, and of steroidogenic enzymes [12]. Since KGN cells are capable of steroidogenesis, it is important to understand the mechanisms through which they regulate the interplay of steroidogenesis and antioxidant defence system, thereby answering the question whether KGN cells can be a true representation of human primary granulosa cells in studies of steroidogenesis. To examine this, we used publicly available RNA sequencing data to compare KGN cells to human primary granulosa cells. We examined the expression of steroidogenic genes, antioxidant genes and the response to treatments with dbcAMP in both KGN cells and in KGN cells in which we knocked down the expression of the antioxidant gene *TXNRD1*.

## Materials and methods

### RNA-sequencing (RNA-seq) analysis

Public RNA-seq data sets GSE161341 (KGN cells; sample GSM4905063) [13], GSE130664 (KGN cells; samples GSM4162518 and GSM4162519) [14] and GSE193123 (primary human granulosa cells from three secondary follicles; samples GSM5773736, GSM5773737 and GSM5773738) [15] were downloaded from the Gene Expression Omnibus (GEO) database. Adapters and low-quality bases (length less than 50 bp and PHRED score less than 20) were removed from the FASTQ files using the AdapterRemoval tool (version: 2.2.2), followed by a quality control check by FastQC (version: 0.11.7). Only FASTQ files that passed quality control were quantified at transcript level by *Kallisto* (version: 0.43.1), using default settings.

For differential gene expression analysis, we used the R package tximport (version: 1.28.0) to summarise *Kallisto* transcript level estimates to gene level with the recommended scaling method lengthScaledTPM for limma-voom pipeline [16, 17]. Using gene level counts, a DGEList object was created by DGEList function in R package edgeR (version: 3.42.4) [18] and filtered with filter-Byexpression function with specifying the group argument. The filtered DGEList object was transformed by voom function from R package limma (version: 3.56.2) [19] and then the voom object was processed with linear fit and Bayes correction to detect differentially-expressed genes (DEGs) which were identified using the criteria |log2(fold change)| > 1 and false-discovery rate (FDR) < 0.05. For gene set enrichment analysis, all remaining genes after filtering were ranked by $\log_2$(fold change) and used as input for the gseGO function in R clusterProfiler package (version: 4.8.2) [20]. Only pathways with FDR < 0.05, |enrichmentScore| > 0.5 and above 40 core enrichment genes were considered significant. The TMM normalised expression counts of all genes in the clean dataset and relevant differential gene expression analysis are listed in S1 Table.

## Cell culture

The human granulosa-like tumour cell line, KGN, was donated by Dr. Carmela Ricciardelli from The University of Adelaide, SA, Australia. To induce steroidogenesis in the KGN cell line, 2–3 x$10^4$ cells (passages: 10–20) were cultured in Dulbecco's modified Eagle's medium / Nutrient Mixture Ham's F-12 (DMEM/F12) (GIBCO/ Life Technologies) supplemented with 10% fetal calf serum (SAFC, Brooklyn, VC, Australia), 100 U/ml penicillin/ 100 mg/ml strepto-mycin and 0.25 μg/ml fungizone (both Thermo-Fisher Scientific, Waltham, MA, USA) at 37˚C, 5% $CO_2$. After 24 h, the cells were cultured for another 48 h with the media as above including 100 nM androstenedione (A4) and were untreated or treated with either 1 mM dbcAMP or 10 μM forskolin. To induce ROS-production, KGN cells were additionally treated with cadmium chloride (2, 5 or 10 μM; Sigma-Aldrich/Merck, St. Louis, MO, USA) for 48 h. Each treatment was repeated five times independently. The concentrations of dbcAMP and forskolin were chosen based on previous reports [1, 8].

## RNA isolation and semi-quantitative real time-PCR

After the 48 h of treatment, the cells were washed with ice-cold phosphate-buffered saline (PBS) two times and directly lysed in the TRIzol™ Reagent (Thermo-Fisher Scientific, Waltham, MA, USA), followed by the RNA isolation procedure according to the manufacturer's instructions. Two μl glycogen (Thermo-Fisher Scientific, Waltham, MA, USA) were used during the isopropanol precipitation step to improve the RNA yield. DNaseI (Promega/Life Technologies Australia Pty Ltd, Tullamarine, Vic, Australia) was also employed to eliminate the potential genomic DNA contamination of the RNA.

One hundred ng of high-quality DNase-treated RNA was used to synthesise cDNA with 250 ng random hexamer primers (Sigma-Aldrich/Merck, St. Louis, MO, USA) and 200 U Superscript III reverse transcriptase (Thermo Fisher Scientific, Waltham, MA, USA), followed by quantitative real time-PCR (qRT-PCR) using Power SYBR™ Green PCR Master Mix (Applied Biosystems, Foster City, CA, USA). Each reaction contained 2 μl 0.5 ng/μl cDNA, 5 μl of Power SYBR™ Green PCR Master Mix, 0.1–0.3 μl each of 10 μM forward and reserve primers for target genes and 2.7–2.9 μl of DEPC-treated water. The amplification conditions were 95˚C for 15 sec, then 60˚C for 60 sec for 40 cycles using a Rotor Gene 6000 cycler (Qseries, Qiagen GmbH, Hilden, Germany). Threshold cycle (Ct) values were determined using Rotor Gene 6000 software at a threshold of 0.05 normalised fluorescent unit. Gene expression was represented by the mean of $2^{-\Delta Ct}$. The primers are listed in S1 Table and each primer pair efficiency was tested as described previously [21].

## Hydrogen peroxide ($H_2O_2$) measurement

Extracellular $H_2O_2$ was measured with the Amplex® Red Hydrogen Peroxidase/Peroxide Assay Kit (Invitrogen/Thermo-Fisher Scientific). In brief, supernatants (50 µl) from cells after 48 h of treatments were mixed with freshly prepared Amplex red working solution containing 100 µM Amplex red dye and 0.2 U/ml horseradish peroxidase. Then, the mixture was incubated at room temperature for 30 min and the fluorescence of the product resorufin (excitation: 535nm, emission: 595 nm) was measured with a plate reader (Dynex Technologies, Chantilly, VA, USA). The standard curve was linear in the range of 0–5 µM $H_2O_2$ and all samples were within that range.

## Immunostaining for DNA damage

To determine early signs of DNA damage, we performed immunofluorescence staining for the phosphorylated H2A histone family member X (γH2Ax), which identifies double-strand breaks in the DNA. For this, 25,000 KGN cells were cultured on coverslips in 24-well plates. After 48 h of treatment, the cells were fixed with 4% paraformaldehyde in PBS at room temperature for 20 min, followed by permeabilisation with 0.1% Triton X-100 in PBS for 25 min at room temperature. To reduce nonspecific binding of antibodies, cells were blocked in 2% bovine serum albumin in PBS for 1 h at room temperature. Then, cells were incubated with rabbit monoclonal Anti-γH2Ax (Sigma-Aldrich/Merck, St. Louis, MO, USA) at 1:1,000 at 4°C overnight. After three washes in PBS, cells were incubated with goat-anti-rabbit Alexa fluor 488 antibody (Abcam, Melbourne, VIC, Australia) at 1:1,000 at room temperature for 1 h and then washed with PBS/ 0.1% Tween 20 three times, each at 5 mins. Then, cell nuclei were counterstained with DAPI (Molecular Probes, Eugene, OR, USA) for 15 min at room temperature. Finally, cells were mounted on glass slides using DAKO mount medium (Agilent, Santa Clara, CA, USA) and imaged with a confocal laser-scanning microscope (FV3000, Olympus, Adelaide, SA, Australia). Image J was used to quantify the number of γH2Ax foci per nucleus in each sample. On average twenty-five to thirty cells per treatment and biological repeat were quantified and then the mean γH2Ax foci for each group was used for statistical analysis. Three biological replicates were performed.

## Silencing of *TXNRD1*

Commercial negative control (Catalog #: 4390843) and *TXNRD1* (Catalog #: 4390824; siRNA ID: s757) siRNAs with Locked nucleic acid (LNA) chemical modifications were purchased from Thermo Fisher Scientific. siRNA transfection was performed following the manual of Lipofectamine™ RNAiMAX Transfection Reagent (Thermo-Fisher Scientific, Waltham, MA, USA). In brief, 20,000 KGN cells were seeded in 24-well plates for 24 h in complete culture medium (see above) without antibiotics. Then, 50 µl lipofectamine-RNAi duplex was added to cells to achieve the 10 nM final concentration of siRNA per well. For lipofectamine control, 50 µl lipofectamine dilution was added to cells. After 6 h incubation, the culture medium containing the siRNA/lipofectamine was removed and cells were then either not treated or incubated with 1 mM dbcAMP for 48 h. Then cells were collected for RNA extraction and qPCR as described above. Each treatment was repeated five times independently.

## Statistical analyses

Results are presented as mean ± SEM. To compare gene expression levels in cultured cells, statistical comparisons were conducted using one-way ANOVA, two-way ANOVA or Welch's t-test as specifically indicated in each figure legend. As one-way or two-way ANOVA were

applied, the differences between each comparison were assessed by Tukey's post-hoc test. All statistical analyses were carried out using both Prism9 (GraphPad Software, La Jolla, CA, USA) and R software.

## Results

### Comparison of gene expression data between KGN and primary granulosa cells

Differential expression analysis using |log2(fold change)| > 1 and FDR < 0.05 revealed 2,220 genes being more highly expressed and 2,823 genes being more lowly expressed in KGN cells versus human granulosa cells (Fig 1A and S2 Table). *TXNRD1* was the only gene of the candidate list of steroidogenic enzymes and antioxidants which was elevated in KGN cells (Fig 1A). Steroidogenic genes *CYP11A1* and *CYP19A1*, their electron transport chain members *FDXR*, *FDX1* and *POR*, as well as the antioxidants *GPX7*, *SOD2* and *TXN2* were all lower in KGN cells

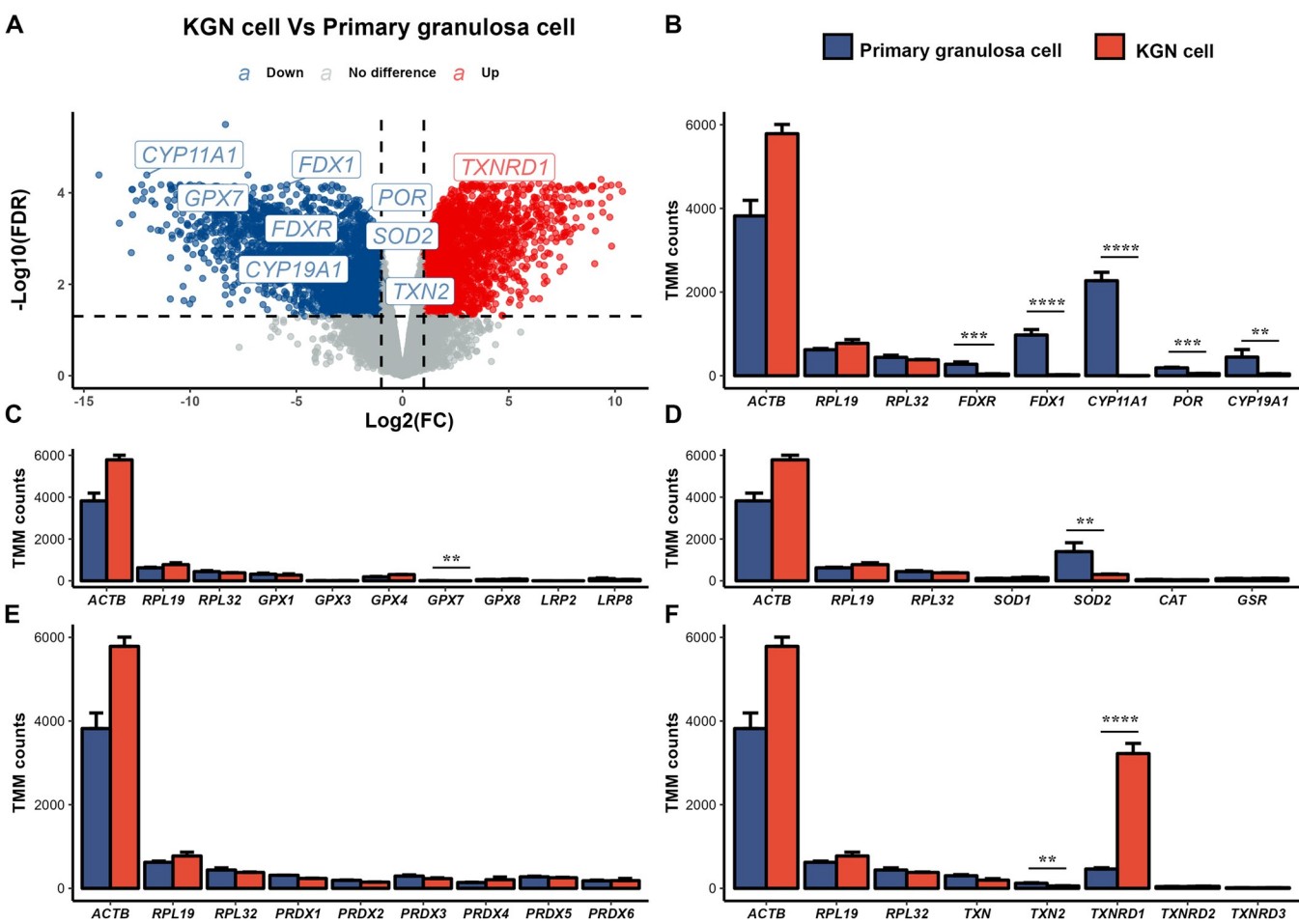

**Fig 1. Comparison of RNA-seq expression data between KGN cells and human primary granulosa cells.** (A) Volcano plot showing differentially expressed genes in KGN versus primary granulosa cells. Upregulated genes (fold change > 2, FDR < 0.05) in KGN cells are shown in red, downregulated genes (fold change < -2, FDR < 0.05) in blue. (B) Expression of steroidogenic genes and electron transport chain components, (C) glutathione peroxidases (*GPX1*, *GPX3*, *GPX4*, *GPX7* and *GPX8*) and selenium uptake receptors (*LRP2*, *LRP8*), (D) glutathione-disulfide reductase (*GSR*), sodium dismutases (*SOD1-2*) and catalase (*CAT*), (E) peroxiredoxins (*PRDX1-6*), (F) thioredoxins (*TXN1-2*) and thioredoxin reductases (*TXNRD1-3*). (B) to (F) each gene expression graph additionally shows the expression of the house-keeping genes *ACTB*, *RPL19* and *RPL32* (n = 3). Data are presented as mean ± SEM. A gene expression of |fold change| > 2 and FDR < 0.05 was considered significant. **$P < 0.01$, ***$P < 0.001$ and ****$P < 0.0001$.

(Fig 1A). Comparing the expression levels in detail using the trimmed mean of M values (TMM) from the RNAseq data revealed that all steroidogenic genes and relevant electron transport chain genes were expressed at significantly higher levels in primary human granulosa cells compared to KGN cells (Fig 1B). In the GPX family, only *GPX7* was differentially expressed and significantly lower in KGN cells, whereas *GPX1*, *GPX3*, *GPX4* and *GPX8* were unchanged (Fig 1C). *GPX2*, *GPX5* and *GPX6* were removed from DGE analysis due to their low expression. Both selenium-uptake receptors, *LRP2* and *LRP8*, were not differentially expressed between primary human granulosa cells and KGN cells (Fig 1C). *SOD2* was also significantly lower expressed in KGN cells (Fig 1D). However, *SOD1*, *CAT* and *GSR* (Fig 1D) as well as the members of the PRDX family (Fig 1E) were unchanged. *TXN2* was significantly lower expressed in KGN cells, whereas *TXN*, *TXNRD2* and *TXNRD3* were unchanged (Fig 1F). Interestingly, *TXNRD1* was not only the only antioxidant gene significantly higher expressed in KGN cells, it was also extremely highly expressed—ten times higher than any other antioxidants in KGN cells and two times higher than *SOD2*, the highest antioxidant in primary human granulosa cells. Gene set enrichment analysis also showed that highly expressed genes and pathways in primary granulosa cells included steroid biosynthetic process and cholesterol metabolic process (Fig 2A), while genes enriched in KGN cells were connected with DNA processing, chromosome segregation and kinetochore pathways (Fig 2B).

## Effect of dbcAMP and forskolin on steroidogenic enzymes in KGN cells

Both dbcAMP and forskolin significantly increased the expression of *CYP11A1* and *CYP19A1* in cultured KGN cells, and the effect appeared stronger with dbcAMP (Fig 3C and 3E). Furthermore, forskolin stimulation significantly reduced the expression of *FDX1* (Fig 3A) and increased *POR* (Fig 3D), whereas dbcAMP had no effect on either gene. The expression of *FDXR* remained unchanged with both treatments (Fig 3B).

## ROS production and DNA damage

Stimulation of KGN cells with dbcAMP or forskolin did not result in changes in the extracellular $H_2O_2$ levels. However, treatment of KGN cells with different doses of cadmium chloride ($CdCl_2$) as positive control clearly induced $H_2O_2$ levels. The treatment with 5 μM and 10 μM $CdCl_2$ caused morphological changes with cells rounding up and detaching (Fig 4A). The levels of $H_2O_2$ were, as expected, significantly increased with 5 μM $CdCl_2$ but not with 2 μM $CdCl_2$ even though an increasing tendency was observed (Fig 4B). However, 10 μM $CdCl_2$ did not show any effect on $H_2O_2$, which might be due to the majority of cells being dead or dying during the 48 h.

One major consequence of excessive ROS accumulation is DNA damage. Therefore, immunostaining for γH2Ax (Fig 4C), an indicator of DNA damage, was carried out and the γH2Ax positive foci per nucleus counted (Fig 4D). A significant increase in the number of γH2Ax foci per nucleus could be observed with both 5 μM and 10 μM $CdCl_2$, whereas a 2 μM concentration did not show a significant effect. Neither dbcAMP nor forskolin treatment resulted in increased γH2Ax foci per nucleus.

## Antioxidant gene expression after steroidogenesis induction

The effects of treatment of KGN cells with forskolin or dbcAMP on the expression of glutathione peroxidases (GPX; Fig 5A–5C), superoxide dismutases (SOD; Fig 5E and 5F), catalase (CAT; Fig 5G), peroxiredoxins (PRDX; Fig 6), glutathione disulphide reductase (GSR; Fig 7A), thioredoxins (TXN; Fig 7B and 7C) and thioredoxin reductases (TXNRD; Fig 7D–7F) were examined. We also analysed the expression of the selenium-uptake receptor 8 (LRP8; Fig 5D),

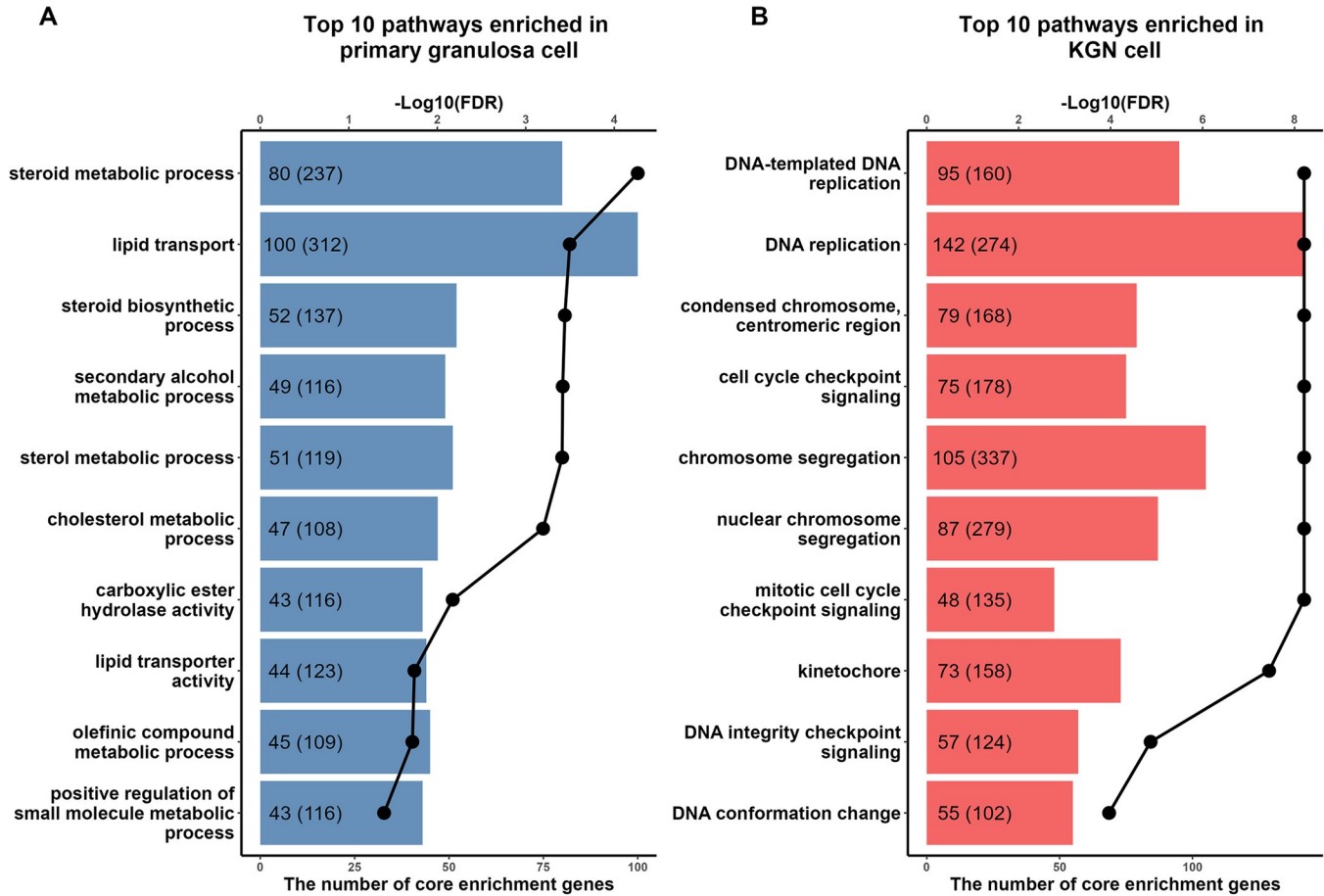

**Fig 2. Geneset enrichment analysis of differentially expressed genes in KGN and primary granulosa cells.** (A) Top 10 differential Gene Ontology (GO) pathways enriched in primary granulosa. (B) Top 10 differential GO pathways enriched in KGN cells. The dotted line represents the FDR of each differential GO pathway. In the bar, the text outside the parentheses represents the number of core enrichment genes that contribute most to the GO pathway. The text inside the parentheses indicates the total number of genes enriched for the pathway in the dataset. Only pathways with false discovery rate (FDR) < 0.05, |enrichmentScore| > 0.5 and > 40 core enrichment genes were considered significant.

which plays a role for selenium-containing antioxidants such as the GPXs. We only present the expression data for *GPX1*, *GPX3* and *GPX8* as the expression of the remaining GPXs (*GPX2*, *4*, *5*, *6* and *7*) was very low to nil (S3 Table). *GPX3* was significantly reduced after forskolin stimulation but unchanged after dbcAMP treatment (Fig 5B). *GPX1* and *GPX8* were not affected by either stimulation (Fig 5A and 5C). The expression of *LRP8* was significantly reduced by treatment with forskolin or dbcAMP (Fig 5D). Treatment with dbcAMP significantly decreased the expression of *SOD2* (Fig 5F) but had no effect on *SOD1* or *CAT* (Fig 5E and 5G). Forskolin treatment did not affect *SOD1*, *SOD2* or *CAT* (Fig 5E–5G). *PRDX2* expression significantly increased with dbcAMP (Fig 6B), whereas *PRDX6* significantly decreased with forskolin treatment (Fig 6F). Both stimulations showed no effect on *PRDX1* (Fig 6A), *PRDX3* (Fig 6C), *PRDX4* (Fig 6D) or *PRDX5* (Fig 6E). Treatment with dbcAMP resulted in significant reduction of *GSR* (Fig 7A) and *TXN* (Fig 7B). Both stimulation treatments significantly reduced *TXNRD1* (Fig 7D). *TXN2* (Fig 7C), *TXNRD2* (Fig 7E) and *TXNRD3* (Fig 7F) were unchanged in their expression. *PRDX1* (Fig 6A), *TXN* (Fig 7B) and *TXNRD1* (Fig 7D) were the highest expressed antioxidant genes in cultured KGN cells.

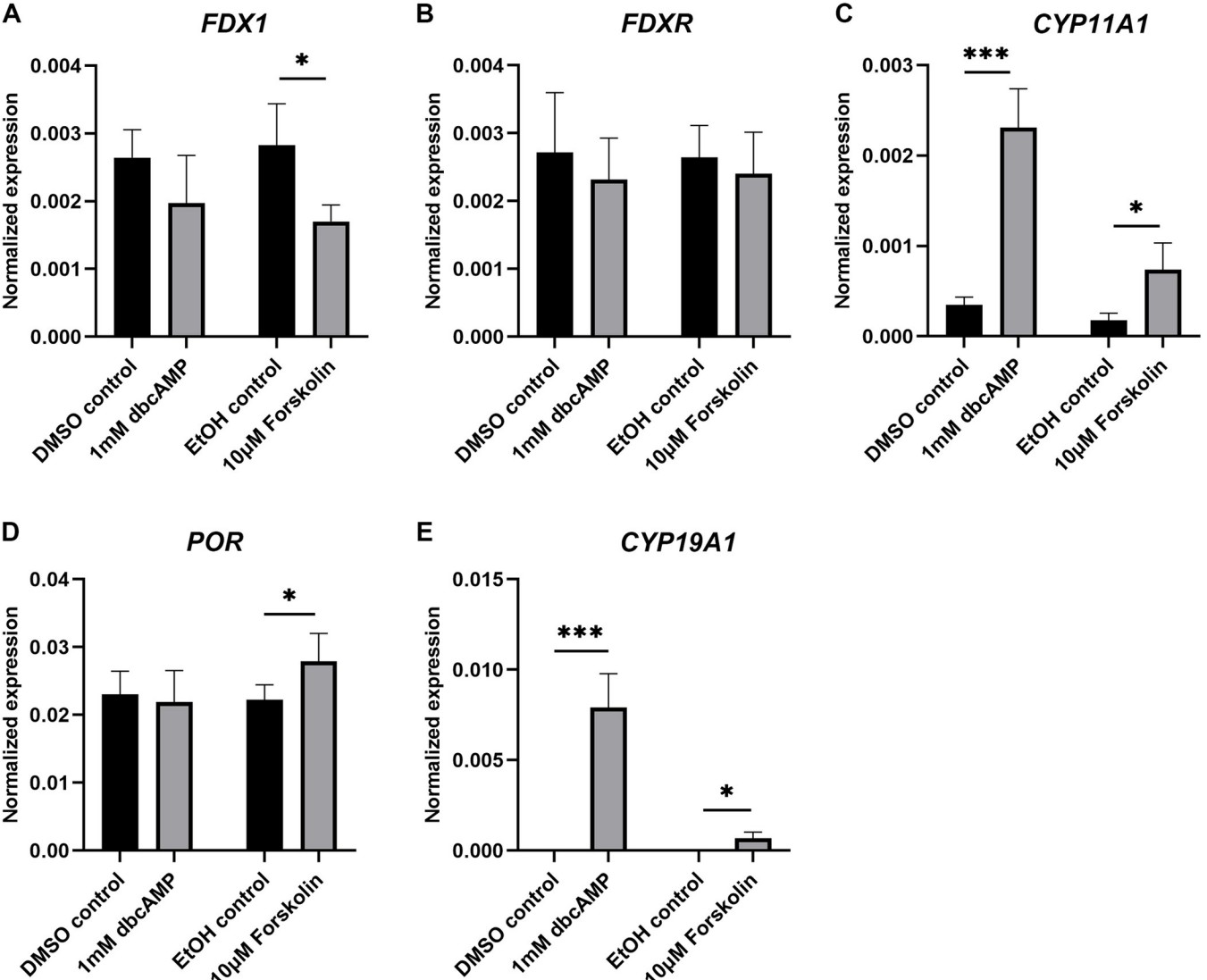

**Fig 3. mRNA expression of steroidogenic enzymes and their electron transport chain components after induction of steroidogenesis in KGN cells *in vitro*.** Cells were treated for 48 h with 1 mM dbcAMP or 10 μM forskolin and then mRNA expression levels determined (n = 5). mRNA expression for *FDX1* (A), *FDXR* (B), *CYP11A1* (C), *POR* (D) and *CYP19A1* (E) were normalised to *GAPDH* expression. Data are presented as mean ± SEM. Welch's t test was used to analyse the pairwise comparison between DMSO control and 1 mM dbcAMP or between ethanol control and 10 μM forskolin. *$P < 0.05$, ***$P < 0.001$.

### Knockdown of *TXNRD1* in KGN cells

*TXNRD1* siRNA transfection of KGN cells resulted in a significant inhibition of *TXNRD1* expression and a knockdown efficiency above 80% (Fig 8D). *TXN2* (Fig 8C), *TXNRD2* (Fig 8E) and *TXNRD3* (Fig 8F) were not changed after knockdown of *TXNRD1*, but a significant increase of *GSR* (Fig 8A) and *TXN* (Fig 8B) was observed in cells with *TXNRD1* knockdown, with or without dbcAMP treatment. Knockdown of *TXNRD1* only significantly increased the expression of *GPX3* (Fig 9B) and *SOD2* (Fig 9F) in the cells without dbcAMP stimulation but significantly reduced the expression of *GPX8* regardless of dbcAMP treatment (Fig 9C). *GPX1* (Fig 9A), *LRP8* (Fig 9D), *SOD1* (Fig 9E) and *CAT* (Fig 9G) were not affected by *TXNRD1* knockdown. Furthermore, *TXNRD1* knockdown showed no effect on *PRDX1* (Fig 10A), *PRDX2* (Fig 10B), *PRDX4* (Fig 10D) and *PRDX5* (Fig 10E), whereas *PRDX3* (Fig 10C) and

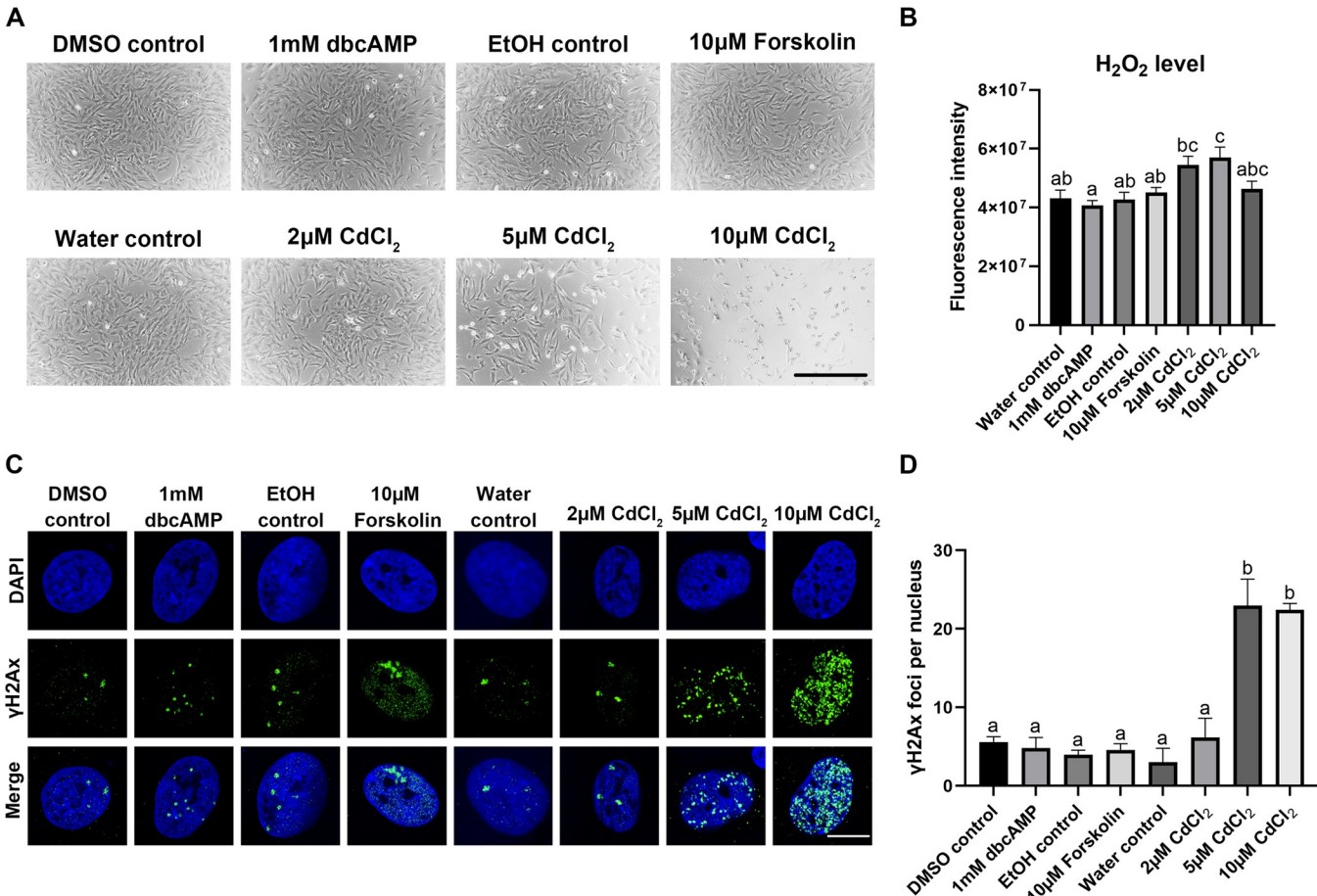

**Fig 4. ROS production and DNA damage in KGN cells *in vitro*.** (A) Representative brightfield images of KGN cells treated with 1 mM dbcAMP, 10 μM forskolin and different doses of cadmium chloride (CdCl₂)after 48 h. Scale bar: 300 μm. (B) $H_2O_2$ levels measured by Amplex Red assay after 48 h of treatment (n = 5). (C) Representative images of DNA damage in KGN cells shown by staining for γH2Ax (green) and counterstain of nuclei with DAPI (blue). Scale bar: 10 μm. (D) Quantification of DNA damage by using Image J (n = 3). One-way ANOVA with Tukey's post-hoc test were used to analyse the data. Bars with different letters are statistically significantly different from each other ($P < 0.05$).

*PRDX6* (Fig 10F) were significantly elevated with *TXNRD1* knockdown in cells with or without dbcAMP stimulation, respectively. Knockdown of *TXNRD1* had no effect on the expression of electron transport chain genes *FDX1* and *FDXR* (Fig 11A and 11B), but resulted in significant inhibition of the stimulatory effect of dbcAMP on *CYP11A1* (Fig 11C) and *CYP19A1* (Fig 11E) expression. *POR* on the other hand was significantly increased (Fig 11D).

## Discussion

Using RNA seq analysis it was discovered that KGN cells, a model of steroidogenic human granulosa cells, expressed very high levels of the antioxidant enzyme *TXNRD1*. Since steroidogenesis produces ROS as a byproduct of the action of the steroidogenic cytochrome P450 enzymes, we explored the interplay in the expression of steroidogenic enzymes, their electron transport chain members and antioxidant genes in KGN cells. We compared the expression of these genes in KGN cells with human primary granulosa cells. The effects on these genes in KGN cells was also examined when the level of expression of *TXNRD1* was reduced using siRNA. We additionally examined ROS production and DNA damage after stimulation of steroidogenesis in KGN cells.

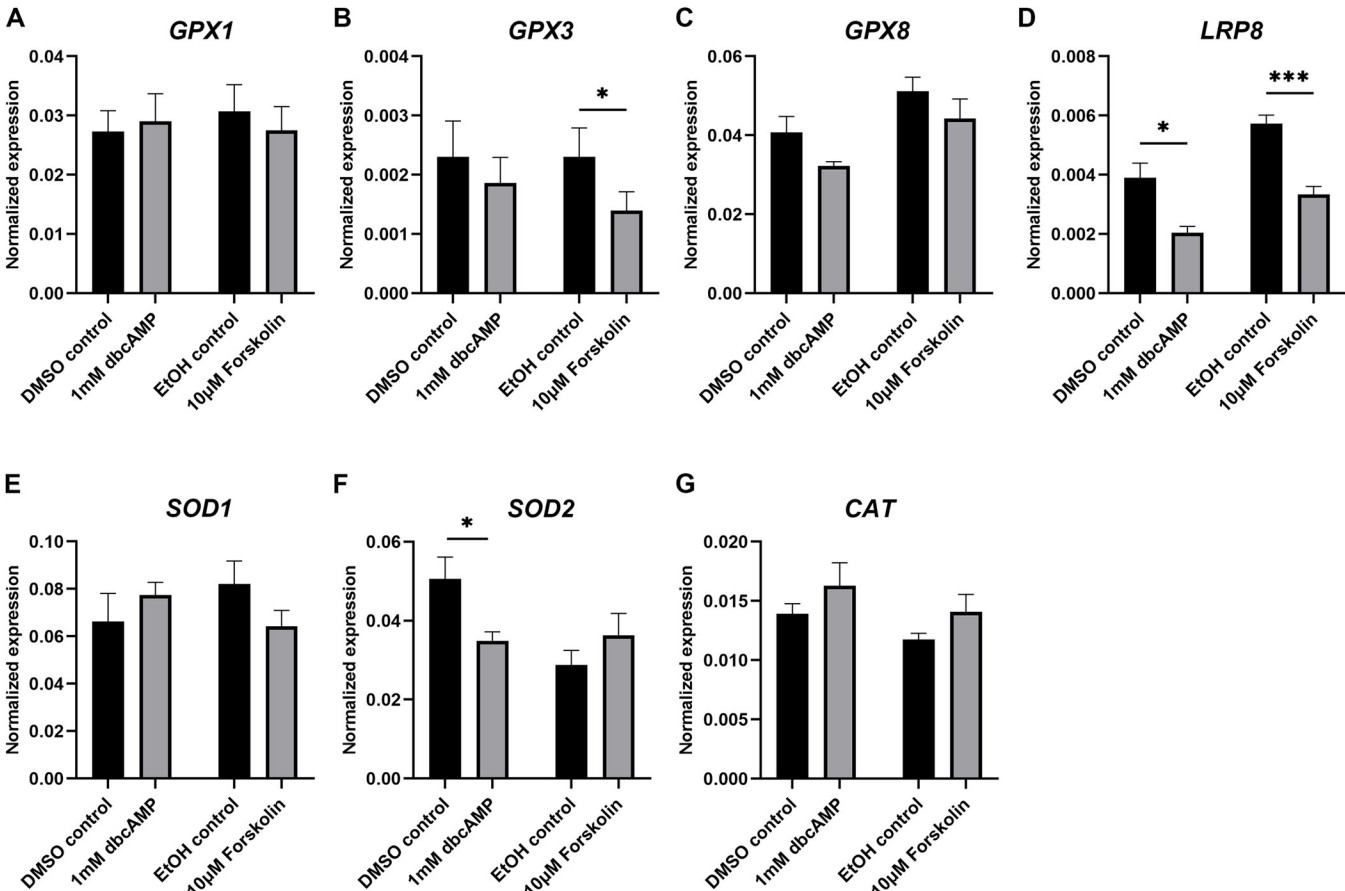

**Fig 5. mRNA expression levels for glutathione peroxidases, selenium-uptake receptor 8, superoxide dismutases and catalase in KGN cells after stimulation with 1 mM dbcAMP or 10 μM forskolin for 48 h.** mRNA expression for *GPX1* (A), *GPX3* (B), *GPX8* (C), *LRP8* (D), *SOD1* (E), *SOD2* (F) and *CAT* (G) was normalised to the expression of *GAPDH* (n = 5). Data are presented as mean ± SEM. Welch's t test was used to analyse the pairwise comparison between DMSO control and 1 mM dbcAMP or between ethanol control and 10 μM forskolin. *$P < 0.05$, ***$P < 0.001$.

Maturation of ovarian follicles is associated with an increased production of oestrogen by granulosa cells until ovulation, after which granulosa cells luteinise and secrete increasing amounts of progesterone. The main steroidogenic enzymes in granulosa cells, CYP11A1 and CYP19A1, rely on their corresponding electron transport chain components for electrons [10]. An earlier study in pig ovaries found that a significant increase in the concentration of P450 enzymes was accompanied by an increase in the concentration of electron transport chain components [22]. In bovine granulosa cells, not only were *CYP11A1* and *CYP19A1* expressed at higher levels in large follicles (>10 mm) compared to granulosa cells from small follicles (4–6 mm), so too were their redox partners, *FDXR* and *POR* [23]. However, in the current study, we found that whilst dbcAMP/forskolin increased expression of *CYP11A1* and *CYP19A1* in KGN cells, *FDX1* and *FDXR* expression was not affected or even negatively so. Moreover, *POR* expression was only stimulated by forskolin but not by dbcAMP. This lack of stimulation of electron transport chain components in KGN cells may be due to the much lower amounts of oestrogen produced by KGN cells [1] compared to primary granulosa cells [4]. Redox partners are generally expressed in excess and therefore an induction of steroidogenic enzymes in KGN cells might still not require an upregulation of *FDXR*, *FDX1* and *POR* to increase enzymatic activities.

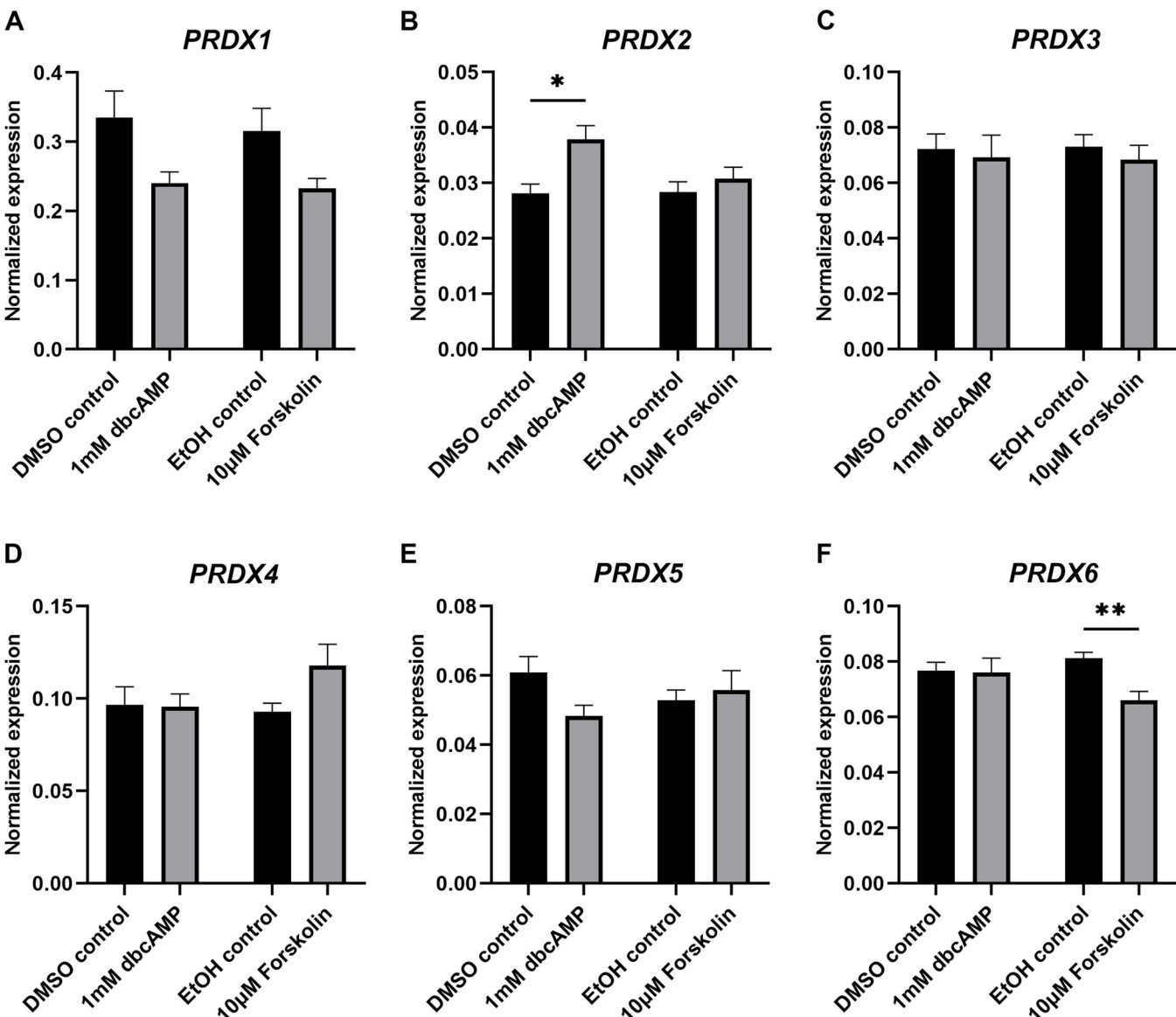

**Fig 6. mRNA expression levels for peroxiredoxins in KGN cells after stimulation with 1 mM dbcAMP or 10 μM forskolin for 48 h.** mRNA expression for *PRDX1* (A), *PRDX2* (B), *PRDX3* (C), *PRDX4* (D), *PRDX5* (E) and *PRDX6* (F) was normalised to the expression of *GAPDH* (n = 5). Data are presented as mean ± SEM. Welch's t test was used to analyse the pairwise comparison between DMSO control and 1 mM dbcAMP or between ethanol control and 10 μM forskolin. *$P < 0.05$, **$P < 0.01$.

Since ROS production due to electron leakage is the unavoidable byproduct of steroidogenesis, steroidogenic cells must have a ROS scavenging system to counteract underlying ROS damage. *TXN* has been shown to be more highly expressed in human granulosa cells from large follicles (> 15 mm) compared to those from smaller follicles (< 15 mm) [24]. Similarly, *GPX1* was more highly expressed in granulosa cells from bovine large follicles (> 10 mm) compared to those from small follicles (4–6 mm) [25]. This suggests that granulosa cells increase the expression of specific antioxidants to counterbalance increased ROS produced by increased oestrogen synthesis. In contrast in KGN cells, we found that the expression of most highly expressed antioxidant genes was unchanged or even downregulated after inducing steroidogenesis, suggesting that KGN cells behave differently to primary granulosa cells. The

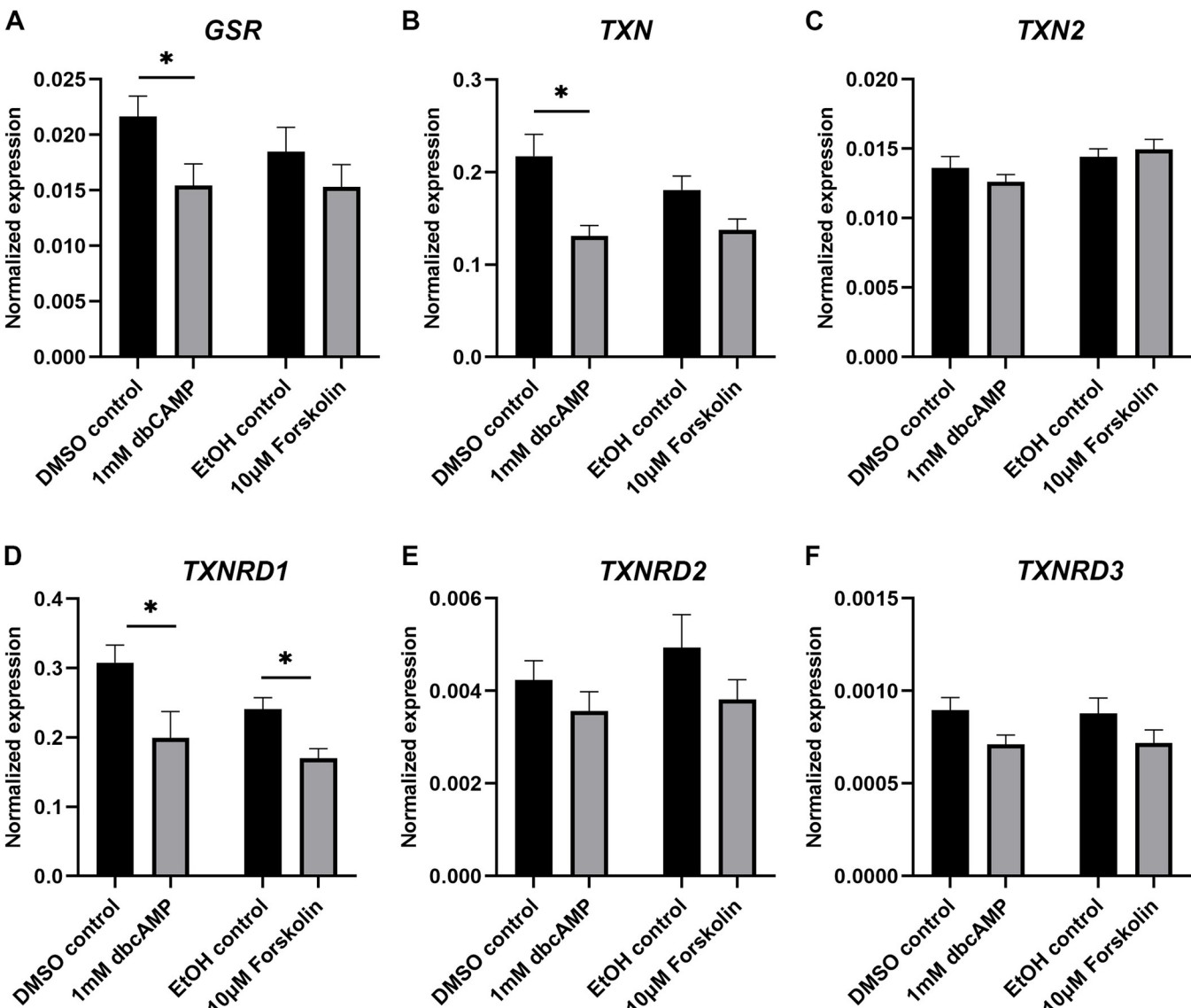

**Fig 7. mRNA expression levels for glutathione-disulfide reductase, thioredoxins and thioredoxin reductases in KGN cells after stimulation with 1 mM dbcAMP or 10 μM forskolin for 48 h.** mRNA expression for *GSR* (A), *TXN* (B), *TXN2* (C), *TXNRD1* (D), *TNXRD2* (E) and *TXNRD3* (F) was normalised to the expression of *GAPDH* (n = 5). Data are presented as mean ± SEM. Welch's t test was used to analyse the pairwise comparison between DMSO control and 1 mM dbcAMP or between ethanol control and 10 μM forskolin. *$P < 0.05$.

expression of the selenium-uptake receptor, *LRP8*, also decreased after stimulation, which could be due to the unchanged or decreased expression levels of selenium-containing GPXs and TXNRDs. Furthermore, we did not find changes in the hydrogen peroxide levels after stimulation of steroidogenesis. Even though *CYP11A1* and *CYP19A1* were significantly increased after dbcAMP and forskolin stimulation, their expression levels are still much lower than reported for primary granulosa cells. Taken together these observations are consistent with the suggestion that any additional ROS produced in the KGN cells under stimulation could still be adequately scavenged by the existing levels of antioxidants.

In terms of antioxidants, we found that the KGN cells had a substantially higher expression of *TXNRD1* compared to primary granulosa cells. With hindsight this is perhaps not

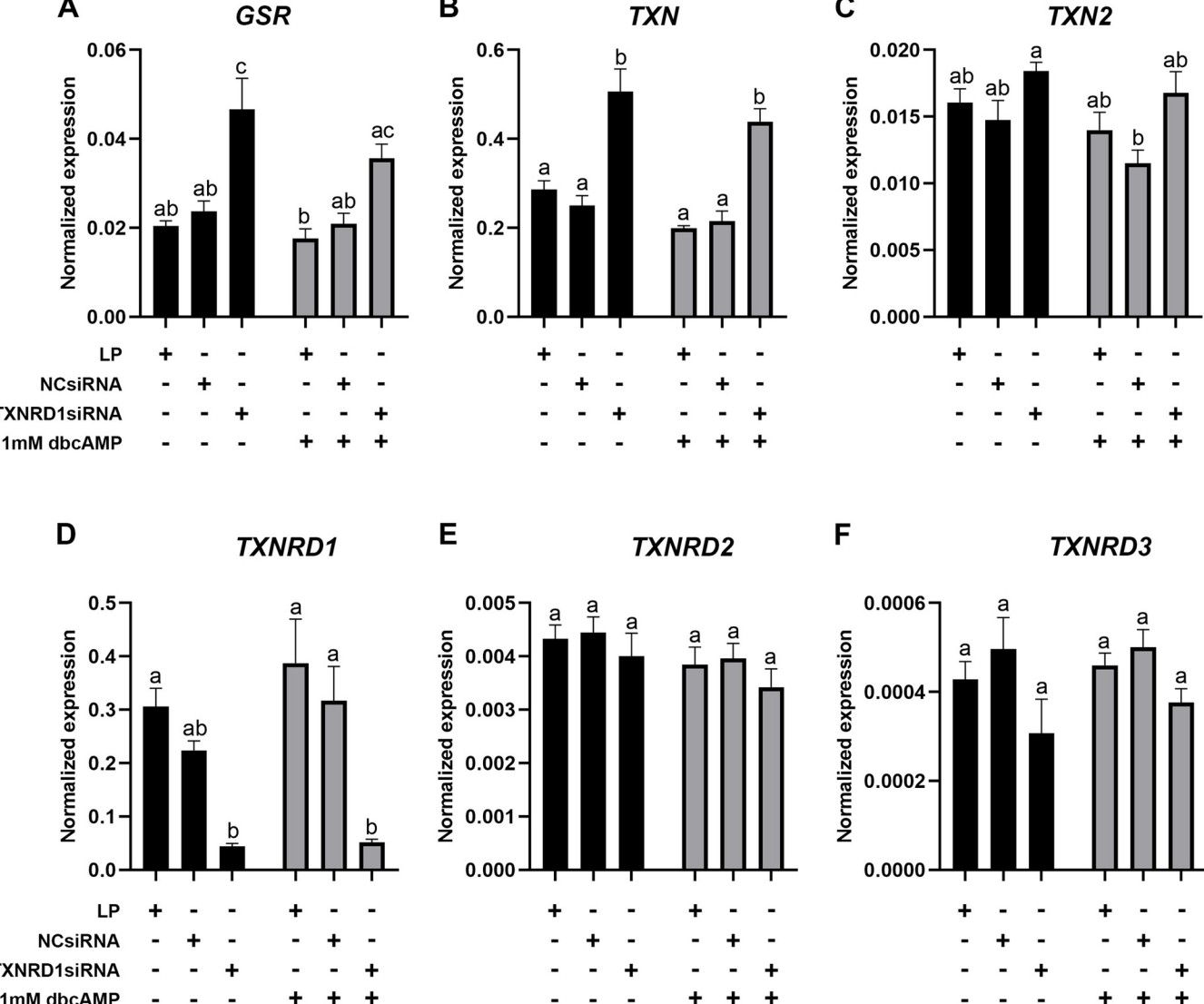

**Fig 8. mRNA expression levels for glutathione-disulfide reductase, thioredoxins and thioredoxin reductases in KGN cells after knockdown of *TXNRD1* followed by 1mM dbcAMP stimulation for 48 h.** mRNA expression for *GSR* (A), *TXN* (B), *TXN2* (C), *TXNRD1* (D), *TNXRD2* (E) and *TXNRD3* (F) was normalised to the expression of *GAPDH* (n = 5). Data are presented as mean ± SEM. Two-way ANOVA with Tukey's post-hoc test was used to analyse the data. Bars with different letters are statistically significantly different from each other ($P < 0.05$). LP–lipofectamine only control; NCsiRNA—negative control siRNA.

surprising as overexpression of thioredoxins and thioredoxin reductases is common in different tumours and is associated with aggressive tumour growth and cancer progression [26–29]. The TXNRD family is NADPH-dependent and mainly located in cytosol and mitochondria [30]. Since TXNRDs are the major reducers of TXNs, they are required for activating TXNs and maintaining the redox homeostasis [31]. For instance, TXNRD1 reduces TXN, which further reduces a number of enzymes, such as PRDXs or methionine sulfoxide reductases for ROS scavenging [32, 33] and ribonucleotide reductase for DNA synthesis [34].

In the current study, we found that an antioxidant defence system consisting of *TXNRD1*, *TXN* and *PRDX1*, showed the highest mRNA expression in KGN cells. Other tumour cell lines, such as A549 [35], FaDu and HeLa cells [36], also express high levels of TXNRD1. Cells

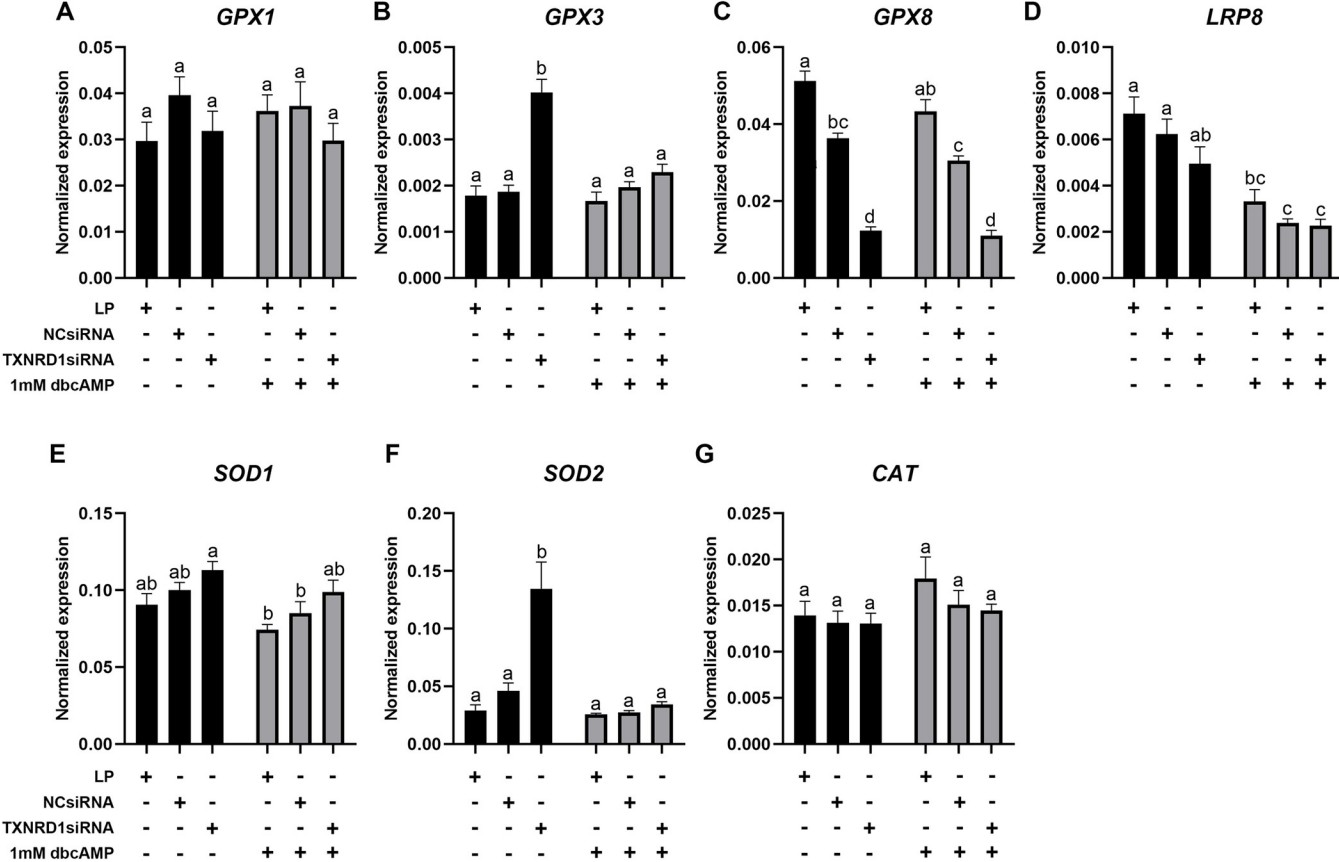

**Fig 9. mRNA expression levels for glutathione peroxidases, selenium-uptake receptor 8, superoxide dismutases and catalase in KGN cells after knockdown of *TXNRD1* followed by 1mM dbcAMP stimulation for 48 h.** mRNA expression for *GPX1* (A), *GPX3* (B), *GPX8* (C), *LRP8* (D), *SOD1* (E), *SOD2* (F) and *CAT* (G) was normalised to the expression of *GAPDH* (n = 5). Data are presented as mean ± SEM. Two-way ANOVA with Tukey's post-hoc test was used to analyse the data. Bars with different letters are statistically significantly different from each other ($P < 0.05$). LP–lipofectamine only control; NCsiRNA —negative control siRNA.

of solid tumours have been shown to not only express high levels of *TXNRD1* but also *GPX4* to avoid oxidative stress and protect against ferroptosis, an iron-mediated cell death (reviewed in [37]). Cancer cells appear to be able to also promote the uptake of selenium by increasing the expression of *LRP8*. TXNRDs, in the presence of glutathione, can reduce inorganic selenium forms, which are then, as selenocysteine, incorporated into selenoproteins such as TXNRDs and GPXs. Surprisingly, in our study we were not able to detect *GPX4* expression in the KGN cells. Interestingly, knockdown of *TXNRD1* in KGN cells did not decrease *TXN* but significantly upregulated its expression, with or without dbcAMP. The possible reason for this may be that TXN can be reduced by other molecules and still act as one of the major antioxidants in *TXNRD1*-deficient cells. Previous studies have shown TXN can also be reduced by the glutathione (GSH)- and glutaredoxin (Grx)-dependent pathways [38, 39]. Knockdown of mitochondrial *Txnrd2* in mouse embryonic fibroblasts resulted in compensatory upregulation of mitochondrial *Grx2*, which is also able to reduce Txn and Txn2 instead of Txnrd2 [40]. We also observed that *GSR*, *GPX3*, *SOD2* and *PRDX6* were significantly increased after *TXNRD1* knockdown in KGN cells without dbcAMP treatment. The increase in *PRDX6* in *TXNRD1*-deficient KGN cells might substitute for *PRDX1* as the major downstream factor of TXN for ROS elimination in KGN cells. Also, the concomitant upregulation of *GPX3* and *SOD2* is

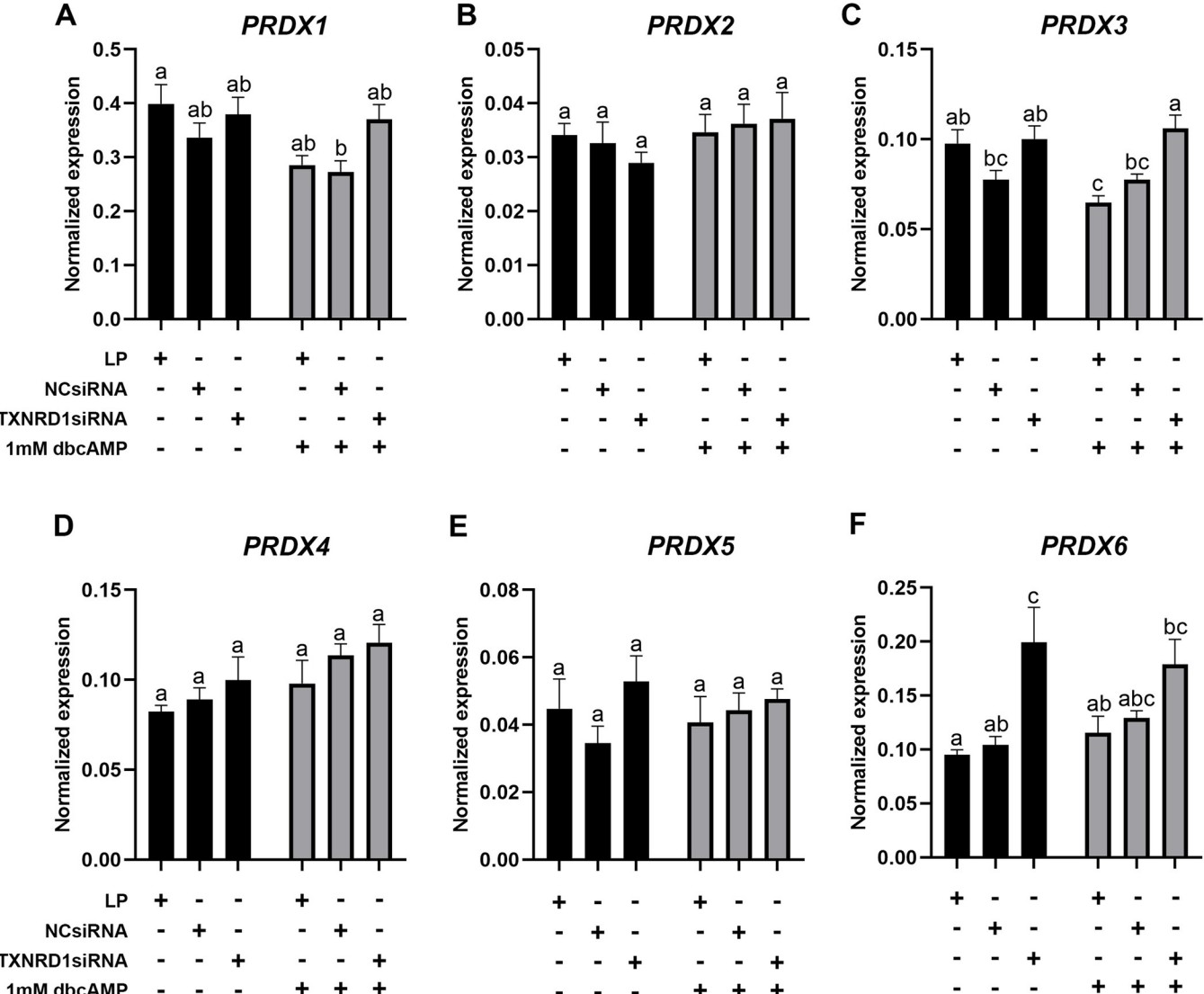

**Fig 10. mRNA expression levels for peroxiredoxins in KGN cells after knockdown of *TXNRD1* followed by 1mM dbcAMP stimulation for 48 h.** mRNA expression for *PRDX1* (A), *PRDX2* (B), *PRDX3* (C), *PRDX4* (D), *PRDX5* (E) and *PRDX6* (F) was normalised to the expression of *GAPDH* (n = 5). Data are presented as mean ± SEM. Two-way ANOVA with Tukey's post-hoc test was used to analyse the data. Bars with different letters are statistically significantly different from each other ($P < 0.05$). LP–lipofectamine only control; NCsiRNA—negative control siRNA.

understandable, since the SOD-GPX system is a classical antioxidant regulatory system, in which SOD converts superoxide radicals into hydrogen peroxide that is subsequently neutralised by GPX by using hydrogen from two GSH molecules resulting in two $H_2O$ molecules and one glutathione disulfide (GSSG) [41, 42]. As GSR regenerates GSH from GSSG [43], the upregulation of *GSR* after knockdown of *TXNRD1* in KGN cells can also be explained.

Interestingly, we observed that knockdown of *TXNRD1* in KGN cells inhibited the stimulation by dbcAMP of *CYP11A1* and *CYP19A1* expression. Similar observations have been made by Zaidi et al., where a deficiency in *Sod2* in mice ovaries resulted in decreased steroidogenesis caused by interference with the cholesterol transport into the mitochondria and downregulation of *Star*, *Cyp11a1*, *Cyp17a1* and *Cyp19a1* [12]. Moreover, exposure to hydrogen peroxide inhibited progesterone production in rat granulosa cells stimulated with LH, cholera toxin,

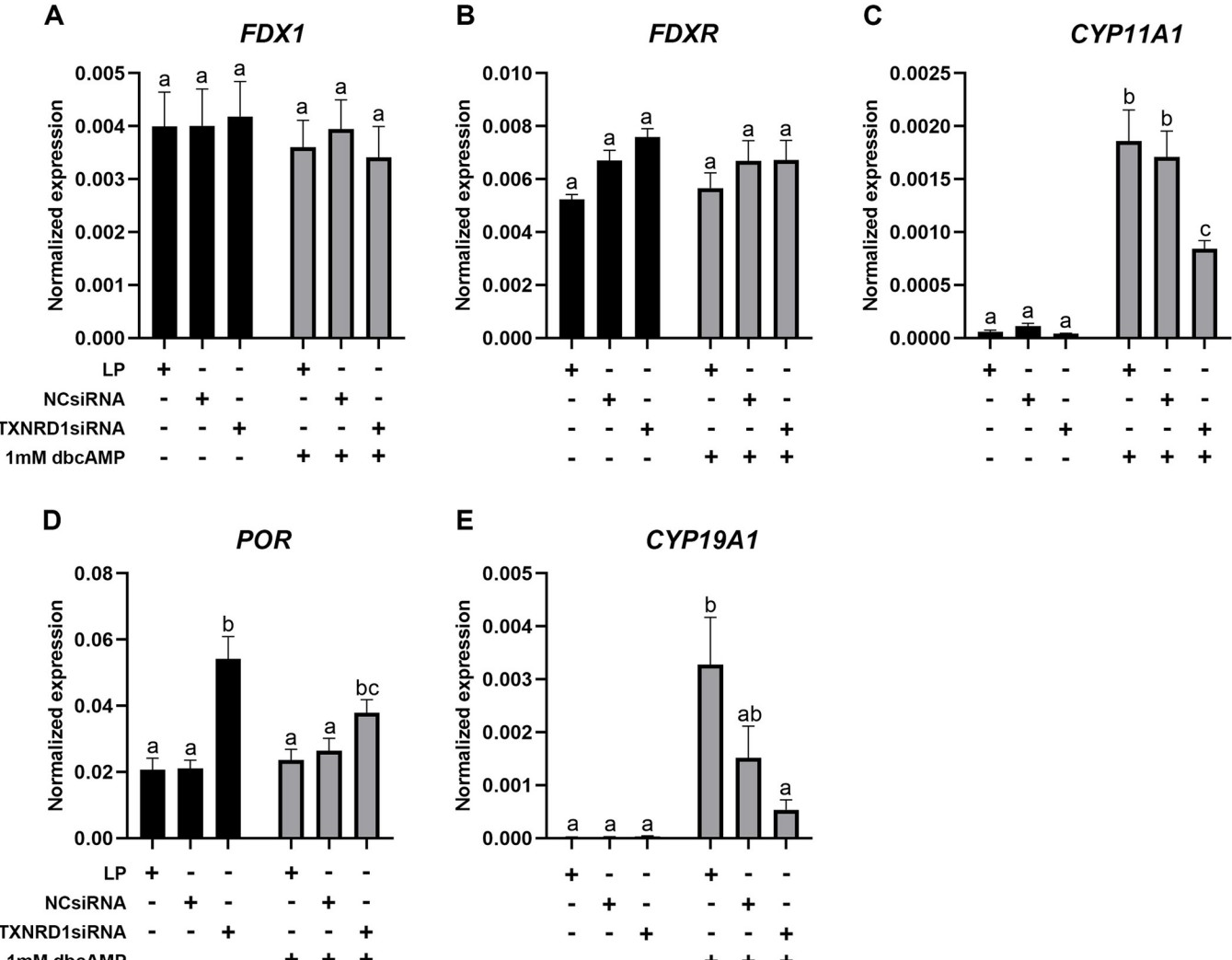

**Fig 11. mRNA expression of steroidogenic enzymes and their electron transport chain components in KGN cells after knockdown of *TXNRD1* followed by 1mM dbcAMP stimulation for 48 h.** mRNA expression for *FDX1* (A), *FDXR* (B), *CYP11A1* (C), *POR* (D), *CYP19A1* (E) was normalised to the expression of *GAPDH* (n = 5). Data are presented as mean ± SEM. Two-way ANOVA with Tukey's post-hoc test was used to analyse the data. Bars with different letters are statistically significantly different from each other ($P < 0.05$). LP–lipofectamine only control; NCsiRNA—negative control siRNA.

forskolin or 8-bromo-cAMP [44]. A similar phenomenon was observed in rat luteal cells upon treatment with xanthine oxidase-generated superoxide [45], lipid hydroperoxide [46] or hydrogen peroxide [47]. In human luteal cells, hydrogen peroxide not only altered hCG-stimulated cAMP accumulation, but also inhibited progesterone and oestrogen synthesis by blocking P450 side-chain cleavage, 3β-hydroxysteroid dehydrogenase, 17β-hydroxysteroid dehydrogenase and P450 aromatase [48]. Collectively, all of these observations suggest that steroidogenic cells, including KGN cells, have a system, which recognises when the antioxidant system is or is likely to be overwhelmed, leading then to a shutdown of steroidogenesis as a protective measure to avoid additional production of ROS.

The mechanism by which the shutdown of steroidogenesis to avoid ROS occurs is under investigation. Knockdown of nuclear factor erythroid 2-related factor 2 (NRF2), the key regulator of cellular antioxidant response, resulted in downregulation of *CYP11A1* in bovine granulosa cells [49]. Aside from inducing TXNs, TXNRDs, sulfiredoxin, PRDXs, GPXs, SOD1,

CAT and several glutathione S-transferases [50], NRF2 also regulates other transcription factor networks such as the sterol regulatory element binding transcription factors (SREBFs/ SREBPs) [49], which is involved in cholesterol biosynthesis. Abidi et al. have shown in the steroidogenic adrenocortical cell line Y1-BS1, that inhibition of steroidogenesis, as a result of excessive oxidative stress (superoxide, hydrogen peroxide, 4-hydroxy-2-nonenal), is mediated by the activated mitogen-activated protein kinase (MAPK) p38 [51]. However, the specific mechanism behind the downregulation of steroidogenic genes after knockdown of *TXNRD1* in KGN cells is not completely understood at this stage.

## Conclusion

In summary, our study revealed that KGN cells are different in their steroidogenic/antioxidant genes profiles to primary granulosa cells. With *TXNRD1* playing such a pivotal role in steroidogenesis in the KGN cells and it being so highly overexpressed, we conclude that KGN cells might not be the most appropriate model of primary granulosa cells for studying the interplay between ovarian steroidogenesis, reactive oxygen species and antioxidants.

## Supporting information

**S1 Table. Primers for quantitative RT-PCR.**
(PDF)

**S2 Table. TMM normalised expression counts of all genes in the RNAseq datasets and relevant differential gene expression analysis results.**
(XLS)

**S3 Table. The normalised expression values (mean ± SEM) of extremely lowly expressed glutathione peroxidases (*GPX2*, *GPX4-7*).**
(PDF)

## Author Contributions

**Conceptualization:** Feng Tang, Katja Hummitzsch, Raymond J. Rodgers.

**Data curation:** Feng Tang.

**Funding acquisition:** Raymond J. Rodgers.

**Investigation:** Feng Tang.

**Methodology:** Feng Tang, Katja Hummitzsch.

**Project administration:** Katja Hummitzsch.

**Resources:** Raymond J. Rodgers.

**Supervision:** Raymond J. Rodgers.

**Writing – original draft:** Feng Tang.

**Writing – review & editing:** Katja Hummitzsch, Raymond J. Rodgers.

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
