## [Decision Letter · Decision Letter 0]

12 Jun 2024

PONE-D-24-15054Unique features of KGN granulosa cells in the regulation of steroidogenic and antioxidant genesPLOS ONE

Dear Dr. Rodgers,

Thank you for submitting your manuscript to PLOS ONE. After careful consideration, we feel that it has merit but does not fully meet PLOS ONE’s publication criteria as it currently stands. Therefore, we invite you to submit a revised version of the manuscript that addresses the points raised during the review process.

We look forward to receiving your revised manuscript.

Kind regards,

Birendra Mishra, DVM, PhD

Academic Editor

PLOS ONE

When you resubmit, please ensure that you provide the correct grant numbers for the awards you received for your study in the ‘Funding Information’ section."

Additional Editor Comments:

Please upload the high-resolution images for the figures.

Reviewers' comments:

Reviewer's Responses to Questions

**Comments to the Author**

1. Is the manuscript technically sound, and do the data support the conclusions?

Reviewer #1: Yes

Reviewer #2: Yes

Reviewer #3: Yes

Reviewer #4: Yes

2. Has the statistical analysis been performed appropriately and rigorously? 

Reviewer #1: Yes

Reviewer #2: Yes

Reviewer #3: Yes

Reviewer #4: Yes

3. Have the authors made all data underlying the findings in their manuscript fully available?

Reviewer #1: Yes

Reviewer #2: Yes

Reviewer #3: Yes

Reviewer #4: Yes

4. Is the manuscript presented in an intelligible fashion and written in standard English?

Reviewer #1: Yes

Reviewer #2: Yes

Reviewer #3: Yes

Reviewer #4: Yes

5. Review Comments to the Author

Reviewer #1: KGN is a widely used cell line, and my team has also used it multiple times. Thank you for the author's research, which has given me a deeper understanding of the application areas and generalizability of KGN results.

Reviewer #2: The article is sound, generally well written and addresses a relevant subject. There are some minor points deserving attention but, in my opinion, the manuscript can be accepted for publication.

In the introduction, within the lines 85-91, after stating the objectives of the study, there are some comments about methodology, some results and even some conclusions which would be more appropriate elsewhere.

The paragraph within the lines 200-204, describes the statistical model, but it does not specify which responses are analyzed.

In the Results session, for the sake of objectivity, I suggest that some methodological comments made at the beginning of each session should be avoided, since the contents were already explained before.

In the Discussion, it would be expected to start the discussion session emphasizing the most relevant implications to the present study. However, the findings of this study are addressed for the first time in the discussion only in line 321, in the middle of the second paragraph.

In the Conclusion, the authors state that “Depending on which aspect a study focuses on, KGN cells might or might not be a good model for primary granulosa cells…”. The comment about KGN cells possibly being a suitable model for primary granulosa cells may somewhat contradict the conclusion mentioned in the abstract, on which is only emphasized that those may not be a good model.

Reviewer #3: Comments to the author

The study by Tang et al. investigates the unique features of KGN granulosa cells in the regulation of steroidogenic and antioxidant genes. More specifically, the authors using public RNA sequencing data compared KGN cells to human granulosa cells and confirmed that KGN cells had extremely higher expression levels of TXNRD1 compared to human granulosa cells, suggesting that overexpression of TXNRD1 is a unique feature of KGN cells. Next, authors studied expression of steroidogenic genes, antioxidant genes and the response to treatments with cAMP in both KGN cells and KGN cells in which authors knocked down the expression of TXNRD1. Overall, this is an interesting work. However, I have nonetheless identified some points to be clarified or resolved. I believe they should revise the manuscript very carefully.

There are some issues that need to be addressed:

1. Line 98, The dataset GSE193123 is the only human granulosa cell dataset used in the study. The authors should provide a detailed description of this dataset, including the physiological status and other relevant information, in an appropriate section.

2. Line 211, There are only three GEO datasets [GSE161341 (KGN cells), GSE130664 (KGN cells) and GSE193123 (primary human granulosa cells from secondary follicle)] in the Material and Method. Please describe the sources and details of the other three datasets (3 KGN datasets with 3 human granulosa cell datasets).

3. Line 236, The study examines the effects of dbcAMP and forskolin on steroid synthase in KGN cells by assessing gene expression levels, not protein levels. The authors should explain why only gene expression was measured and whether it is a more suitable indicator of steroid synthase activity. Consider including additional experiments to support your conclusions.

4. Line 263, The authors found that the expression of some antioxidant genes changed significantly after steroid production induction, such as the decrease of SOD2. Please discuss whether it is necessary to measure changes in SOD activity.

5. Line 109/113/203/204, The used software version should be specified in detail.

6. Line 425, Gene symbol needs italics.

7. Line 590/597, FDR suggests writing it as a full spelling name.

8. The format and font of the Reference list need to be re-checked and modified by the author.

Reviewer #4: 1、in this study, ACTB was used as housekeeping gene. did you confirm the stability in KGN？

2、in Fig.1, the expression levels of all these genes were significant lower than ACTB, is that suitable?

3、How did you determine the concentration of additives （dbcAMP\\Forskolin）?

4、mRNA expression cannot fully reflect the true changes within cells, it is recommended to supplement protein level testing

6. PLOS authors have the option to publish the peer review history of their article (what does this mean?). If published, this will include your full peer review and any attached files.

Reviewer #1: No

Reviewer #2: No

Reviewer #3: No

Reviewer #4: No

---

## [Author Response · Author response to Decision Letter 0]

26 Jun 2024

Reviewer 1 

COMMENT: KGN is a widely used cell line, and my team has also used it multiple times. Thank you for the author's research, which has given me a deeper understanding of the application areas and generalizability of KGN results.

REPLY: Thank you for your recognition of this issue.

Reviewer 2

COMMENT: In the introduction, within the lines 85-91, after stating the objectives of the study, there are some comments about methodology, some results and even some conclusions which would be more appropriate elsewhere.

REPLY: We have rewritten the end of the Introduction now. We say ‘….To examine this we used public RNA sequencing data to compared KGN cells to human granulosa cells. We examined the expression of steroidogenic genes, antioxidant genes and the response to treatments with cAMP in both KGN cells and in KGN cells in which we knocked down the expression of the antioxidant gene TXNRD1.’

COMMENT: The paragraph within the lines 200-204, describes the statistical model, but it does not specify which responses are analyzed.

REPLY: We now indicated that it was to examine gene expression levels. 

COMMENT: In the Results session, for the sake of objectivity, I suggest that some methodological comments made at the beginning of each session should be avoided, since the contents were already explained before.

REPLY: We have done this now. 

COMMENT: In the Discussion, it would be expected to start the discussion session emphasizing the most relevant implications to the present study. However, the findings of this study are addressed for the first time in the discussion only in line 321, in the middle of the second paragraph.

REPLY: We have rewritten the first paragraph of the Discussion so as to inform the reader of what to expect in the remainder of the Discussion. We now say ‘Using RNA seq analysis it was discovered that KGN cells, a model of steroidogenic human granulosa cells, expressed very high levels of the antioxidant enzyme TXNRD1. Since steroidogenesis produces reactive oxygen species as a byproduct of the action of the steroidogenic cytochrome P450 enzymes we explored the interplay in the expression of steroidogenic enzymes, their electron transport chain members and antioxidant genes in KGN cells. We compared the expression of these genes in KGN cells with human primary granulosa cells. The effects on these genes in KGN cells was also examined when the level of expression of TXNRD1 was reduced using siRNA. We additionally examined ROS production and DNA damage after stimulation of steroidogenesis in KGN cells.’

COMMENT: In the Conclusion, the authors state that “Depending on which aspect a study focuses on, KGN cells might or might not be a good model for primary granulosa cells…”. The comment about KGN cells possibly being a suitable model for primary granulosa cells may somewhat contradict the conclusion mentioned in the abstract, on which is only emphasized that those may not be a good model.

REPLY: We have now refocused the Conclusion. We now say ‘With TXNRD1 playing such a pivotal role in steroidogenesis in the KGN cells and it being so highly overexpressed, we conclude that KGN cells might not be the most appropriate model of granulosa cells for studying the interplay between ovarian steroidogenesis, reactive oxygen species and antioxidants.’ This is now in agreement with the abstract. Its simple and accurate.

Reviewer 3

COMMENT: Line 98, The dataset GSE193123 is the only human granulosa cell dataset used in the study. The authors should provide a detailed description of this dataset, including the physiological status and other relevant information, in an appropriate section.

REPLY: We now also provide the GSM numbers and the references from which the data come. We now say ‘Public RNA-seq data sets GSE161341 (KGN cells; sample GSM4905063) [19], GSE130664 (KGN cells; samples GSM4162518 and GSM4162519) [20] and GSE193123 (three primary human granulosa cells from three secondary follicles; samples GSM5773736, GSM5773737 and GSM5773738) [21]’

COMMENT: Line 211, There are only three GEO datasets [GSE161341 (KGN cells), GSE130664 (KGN cells) and GSE193123 (primary human granulosa cells from secondary follicle)] in the Material and Method. Please describe the sources and details of the other three datasets (3 KGN datasets with 3 human granulosa cell datasets).

REPLY: This has been addressed now in the Material and Methods with details in the REPLY immediately above. 

COMMENT: Line 236, The study examines the effects of dbcAMP and forskolin on steroid synthase in KGN cells by assessing gene expression levels, not protein levels. The authors should explain why only gene expression was measured and whether it is a more suitable indicator of steroid synthase activity. Consider including additional experiments to support your conclusions.

REPLY: There are many previous publications with KGN cells showing that the stimulations we used result in an increase in expression of steroidogenic enzymes, resulting in more enzymes at the protein level, more enzymatic activity and more steroids. We have cited their publications. 

Our key finding is that knock down of TXNRD1 expression inhibits -the induction of mRNA for steroidogenic enzymes. If you prevent mRNA being produced in the first place you do not have any protein. No one would expect that if you had blocked the induction of the expression of the mRNA in the first place that you would need to show that you have no protein. Hence, we did not consider conducting such experiments. 

COMMENT: Line 263, The authors found that the expression of some antioxidant genes changed significantly after steroid production induction, such as the decrease of SOD2. Please discuss whether it is necessary to measure changes in SOD activity.

REPLY: The analyses we conducted examined many genes. We followed up on what we considered the most important which was the TXDRD1. In the future SOD could be further examined.

COMMENT: Line 109/113/203/204, The used software version should be specified in detail.

REPLY: Done.

COMMENT: Line 425, Gene symbol needs italics.

REPLY: Done.

COMMENT: Line 590/597, FDR suggests writing it as a full spelling name.

REPLY: Done.

COMMENT: The format and font of the Reference list need to be re-checked and modified by the author.

REPLY: Done.

Reviewer 4

COMMENT: in this study, ACTB was used as housekeeping gene. did you confirm the stability in KGN?

REPLY: Figure 1 shows the levels of three housekeeping genes (ACTB, RPL19, RPL32) in KGN cells and granulosa cells and the expression of each gene is similar between the two cell types.

COMMENT: in Fig.1, the expression levels of all these genes were significant lower than ACTB, is that suitable?

REPLY: Yes, that’s reasonable as ACTB is the major protein involved in the cytoskeleton construction, which means it should be expressed more highly than the other housekeeping genes. This is well known.

COMMENT: How did you determine the concentration of additives （dbcAMP\\Forskolin）?

REPLY: These are standard maximal concentrations used in granulosa cells of many different species and for many decades now, including KGN cells and relevant references are cited [1, 2].

1. Nishi Y, Yanase T, Mu Y, Oba K, Ichino I, Saito M, et al. Establishment and characterization of a steroidogenic human granulosa-like tumor cell line, KGN, that expresses functional follicle-stimulating hormone receptor. Endocrinology. 2001;142(1):437-45. Epub 2001/01/06. https://10.1210/endo.142.1.7862. PMID: 11145608

2. Tremblay PG, Sirard MA. Gene analysis of major signaling pathways regulated by gonadotropins in human ovarian granulosa tumor cells (KGN)dagger. Biol Reprod. 2020;103(3):583-98. Epub 2020/05/20. https://10.1093/biolre/ioaa079. PMID: 32427331

COMMENT: mRNA expression cannot fully reflect the true changes within cells, it is recommended to supplement protein level testing.

REPLY: Please see our response to the third comment by Referee 3.

---

## [Decision Letter · Decision Letter 1]

18 Jul 2024

Unique features of KGN granulosa-like tumour cells in the regulation of steroidogenic and antioxidant genes

PONE-D-24-15054R1

Dear Dr. Rodgers,

We’re pleased to inform you that your manuscript has been judged scientifically suitable for publication and will be formally accepted for publication once it meets all outstanding technical requirements.

Kind regards,

Birendra Mishra, DVM, PhD

Academic Editor

PLOS ONE

Additional Editor Comments (optional):

Reviewers' comments:

Reviewer's Responses to Questions

**Comments to the Author**

1. If the authors have adequately addressed your comments raised in a previous round of review and you feel that this manuscript is now acceptable for publication, you may indicate that here to bypass the “Comments to the Author” section, enter your conflict of interest statement in the “Confidential to Editor” section, and submit your "Accept" recommendation.

Reviewer #3: All comments have been addressed

Reviewer #4: All comments have been addressed

2. Is the manuscript technically sound, and do the data support the conclusions?

Reviewer #3: Yes

Reviewer #4: Yes

3. Has the statistical analysis been performed appropriately and rigorously? 

Reviewer #3: Yes

Reviewer #4: Yes

4. Have the authors made all data underlying the findings in their manuscript fully available?

Reviewer #3: Yes

Reviewer #4: Yes

5. Is the manuscript presented in an intelligible fashion and written in standard English?

Reviewer #3: Yes

Reviewer #4: Yes

6. Review Comments to the Author

Reviewer #3: (No Response)

Reviewer #4: The author made careful revisions based on the suggested changes, so I suggest accept this manuscript and publish online.

7. PLOS authors have the option to publish the peer review history of their article (what does this mean?). If published, this will include your full peer review and any attached files.

Reviewer #3: No

Reviewer #4: No

---

## [Editor Report · Acceptance letter]

30 Jul 2024

PONE-D-24-15054R1 

PLOS ONE

Dear Dr. Rodgers, 

I'm pleased to inform you that your manuscript has been deemed suitable for publication in PLOS ONE. Congratulations! Your manuscript is now being handed over to our production team.

Kind regards, 

on behalf of

Dr. Birendra Mishra 

Academic Editor

PLOS ONE